Silicon mitigates the adverse effects of drought on Lolium perenne physiological, morphometric and anatomical characters

Mastalerczuk Grażyna 1 grazyna_mastalerczuk@sggw.edu.pl
Borawska-Jarmułowicz Barbara 1
http://orcid.org/0000-0002-5234-2008 Sujkowska-Rybkowska Marzena 2
Bederska-Błaszczyk Magdalena 3
Borucki Wojciech 2
http://orcid.org/0000-0002-2867-8839 Dąbrowski Piotr 4 piotr.andrzej.dabrowski@gmail.com
1 Department of Agronomy, Institute of Agriculture, Warsaw University of Life Sciences - SGGW , Warsaw , Poland
2 Department of Botany, Institute of Biology, Warsaw University of Life Sciences-SGGW , Warsaw , Poland
3 Polish Academy of Sciences Botanical Garden–Center for Biological Diversity Conservation in Warsaw-Powsin , Poland
4 Department of Environmental Management, Institute of Environmental Engineering, Warsaw University of Life Sciences-SGGW , Warsaw , Poland
Serim Ahmet Tansel
Electronic publication date: 2025 Feb 12
Publication date: 2025
Volume: 13
Electronic Location ID: e18944
Received 2024 Oct 25; Accepted 2025 Jan 16
Copyright: © 2025 Mastalerczuk et al.
Copyright year: 2025
Copyright holder: Mastalerczuk et al.
License: This is an open access article distributed under the terms of the Creative Commons Attribution License, which permits unrestricted use, distribution, reproduction and adaptation in any medium and for any purpose provided that it is properly attributed. For attribution, the original author(s), title, publication source (PeerJ) and either DOI or URL of the article must be cited.
License URL: https://creativecommons.org/licenses/by/4.0/

Keywords: Chlorophyll a fluorescence, Drought, Leaf anatomy, Perennial ryegrass, Roots, Silicon

Funding: The authors received no funding for this work.

==============================
Limited water resources and natural drought may result in reduced water availability for the population’s needs and the maintenance of the proper vegetation condition. Understanding the impact of drought on turfgrass species is essential to developing strategies that enhance the adaptability of plants to drought stress. It is vital for maintaining green areas in cities under changing climatic conditions. Therefore, studies on the ability of turfgrasses to recover after periods of drought without irrigation are becoming increasingly essential. We conducted research to determine the possibility of reducing the negative impact of drought stress on the photosynthetic efficiency, the morphometric features of plant shoots and roots, and the distribution of biomass of Lolium perenne lawn cultivars in the initial period of growth by applying biostimulant with silicon. We also investigated how drought and silicon (Si) application affect the leaf and root anatomical structure of L. perenne plants. Studies on the influence of drought on the physiological, biometric parameters and anatomical characteristics of two L. perenne cultivars were carried out under two levels of soil moisture (well-watered plants—control and drought caused by the cessation of watering) and also two variants of Si application (with and without Si application). Plants were exposed to drought in the tillering phase for 21 days. After this time, all plants were provided with optimal soil moisture conditions for the next 14 days (recovery period). Measurements of physiological parameters and biometric features of plants were evaluated in four terms: after 7, 14 and 21 days of drought and after recovery. Drought stress in L. perenne cvs. resulted in decreasing values of physiological parameters, especially maximal fluorescence, the quantum efficiency of photosystem II and photosynthetic electron transport rate compared with the values of features in well-watered plants. These adverse impacts were reflected in decreased biomass-related traits: shoot number, shoots and roots biomass, as well as area and length of roots. The application of Si reduced the detrimental effects of drought by accelerating plant growth after stress and increasing the values of most of the investigated physiological parameters. Under drought stress conditions, Si application led to the development of two-cell-layer exodermis, which reduced the water losses in L. perenne roots and shoots and, as a result, improved the drought tolerance of plants.

Introduction

Periodic droughts recurring in recent years in Central Europe (including Poland) have caused large losses in plant production (Łabędzki & Bąk, 2017; Trnka et al., 2020). The increase in the frequency of droughts is primarily the result of an increase in air temperature during the growing season and the occurrence of periods without precipitation or periods with repeated rainfall lower than average. Water deficiency causes deterioration of growth and development conditions and decreases the production of plant biomass (Górski, Kozyra & Doroszewski, 2008). Under projected water deficits, a severe reduction in grassland productivity (Smith, 2011) and ecosystem functions (Jentsch et al., 2011) is expected. The health benefits of turf areas in parks and other urban green spaces have been highlighted in recent years (Neale et al., 2020). During extended droughts, there may be less water available for lawn irrigation due to water shortages and growing human demands for water (Milesi, Elvidge & Nemani, 2009; Gelernter et al., 2015). Studies on the ability of turfgrasses to recover from drought without irrigation are therefore of growing importance (Braun et al., 2022).

Lolium perenne is the most important forage grass cultivated in the temperate climate zone. This species provides high yields with a good forage value for ruminants (Chapman et al., 2022). It is also appreciated for its turf properties. L. perenne germinates quickly and tolerates intensive management. However, it requires a relatively large amount of water to maintain growth (Robins & Bushman, 2023). It belongs to shallow-rooting grasses with a limited drought tolerance (Sampoux et al., 2011).

The way that plants react to drought is a complicated process that differs greatly based on the species or cultivar, the stage of plant development, the severity and duration of stress, and other factors (Lisar et al., 2012). Under conditions of water deficit, plants lose turgor and reduce transpiration by closing their stomata to prevent excessive water loss. At the same time, the uptake of carbon dioxide is hindered, which results in a decrease in the intensity of photosynthesis (Farooq et al., 2009). One technique for examining how the light-dependent phase of photosynthesis responds to abiotic stress is to measure the physiological state of the photosynthetic process by measuring the chlorophyll fluorescence (λmax = 685 nm), which is present in 95% of photosystem II (PSII). Additionally, it provides an indirect estimate of the tissue’s chlorophyll content. Stress causes alterations in the levels of chlorophyll a fluorescence and a decrease in the photochemical efficiency of PSII (Kalaji et al., 2014). The shortage of water hinders the effective use of the absorbed light energy, which is manifested by its increased dissipation by heat and increased fluorescence (Kalaji et al., 2017).

The roots of many plants experience drought stress first; however, they continue to grow while the growth of the shoots is slowed down (Spollen & Sharp, 1991). Lateral root growth is greatly inhibited during drought, primarily due to the inhibition of lateral root meristem activation (Deak & Malamy, 2005). Adaptive characteristics for enduring drought stress include the existence of specialized tissues in plant roots, such as rhizodermis, with a thicker outer cell wall or suberized exodermis, as well as a decrease in the number of cortical layers (Basu et al., 2016). An important trait of perennial grasses is the ability to regrow after a period of drought. As water is lost in plant organs, metabolic processes are gradually inhibited, and when water is reabsorbed, the organs are quickly hydrated and metabolic processes are activated. The level of dehydration tolerance shows variations in the ability of grass species and cultivars to withstand stress, but it can also reveal the potential for plant adaptation (Staniak & Kocoń, 2015). Leaf buds are an essential organ that determines the survival of periods of drought in grasses. They can withstand lower osmotic potential values than a fully grown leaf blade, and they start plant regrowth. The dying of leaves during a prolonged drought allows the movement of proteins, fats and other macromolecules just to buds of young leaves, flowers or seeds, and thus, the plant can regenerate after the stress is over (Volaire, 2003). Recently, there has been a growing interest in the use of elements in plant production that positively affect plant growth, but are not included in either macro- or micronutrients. One of them is silicon (Si). It is the second most abundant element in the lithosphere, after oxygen (Elsokkary, 2018). The silicon is taken up by plants from the soil solution in the form of monosilicic (Si[OH4]) and orthosilicic (H4SiO4) acids, commonly found at concentrations ranging from 0.1 to 0.6 mM at the pH levels found in most agricultural soils (Knight & Kinrade, 2001). Apart from the pH of the soil, Si bioavailability is mostly determined by parameters such as soil texture, temperature, organic matter, and accompanying ions (Tripathi et al., 2021). Application of silicon to the soil can rapidly increase the concentration of orthosilicic acid in the soil solution and can, therefore, be a major treatment in intensive agriculture, especially in soils that are characterized by a low content of soluble silicon (Schaller et al., 2021). Some authors report that this element is mostly or even completely taken up by the roots (Ma & Yamaji, 2006; Guével, Menzies & Bélanger, 2007), while others indicate a high effectiveness of foliar Si application (Sattar et al., 2020; Othman et al., 2021). Plants can be categorized based on silicon (Si) content in tissues as accumulators (e.g., rice, wheat, maize, and sorghum), intermediates (e.g., cucumber, bitter gourd, and melon), or excluders (e.g., tomato, potato, canola, and lentil). The differences are attributed to the different modes of Si uptake (active, passive, and rejective) (Sacała, 2009; Luyckx et al., 2017). Furthermore, these differences primarily due to the ability of various plant species’ roots to absorb Si, which is linked to the expression and function of Si transporters (Souri et al., 2021). In recent decades, there has been a notable increase in interest in the foliar application of silicon as a fertilizer. The effectivity of foliar fertilization is influenced by various factors like plant physiology, species-specific properties (e.g., leaf structure and form), abiotic as well as biotic stressors, spray application, and physico-chemistry of the sprayed solution (Puppe & Sommer, 2018). Silicon may be regarded as an almost necessary element for plants since its deficiencies can result in many problems in plant growth, development, and reproduction (Epstein & Bloom, 2005; Mir et al., 2022). Many studies have shown that the Si application has a positive effect on yielding and also reduces the negative effects of abiotic and biotic stresses of various plant species (Ma, 2004; Epstein & Bloom, 2005; Artyszak, 2018; Siddiqui et al., 2020; Krzymińska, 2021; Mastalerczuk, Borawska-Jarmułowicz & Darkalt, 2023). The influence of Si on physiological parameters greatly depends on plant species, genotype, growth stage, and stress severity (Thorne, Hartley & Maathuis, 2020). Under drought conditions, Si positive effects photosynthesis and chlorophyll levels (Maghsoudi, Emam & Pessarakli, 2016). Additionally, Si consistently lowers oxidative damage across a range of crops and environmental conditions, which is correlated with a notable rise in antioxidant enzyme activity (Sattar et al., 2019). Numerous article provide evidence that Si fertilizers change the plant’s water status during drought, which is partly a consequence of Si increasing water use efficiency (WUE) (Hajiboland, Cheraghvareh & Poschenrieder, 2017). However, the mechanism(s) of this process are still under investigation.

The majority of research focuses on a variety of drought-related characteristics of plants, but they often are related to the sections of the plants that are only above or below ground. This complicates comparisons between studies on grass’s drought tolerance. In this study, the simultaneous response of plant shoots and roots to drought and Si application was assessed. The objectives of the research were: (i) to determine the possibility of reducing the negative impact of drought stress on the photosynthetic efficiency, the morphometric features of plant shoots and roots, and the distribution of biomass of L. perenne lawn cultivars in the initial period of growth, by applying biostimulant with silicon; (ii) to learn how drought and Si application affect leaf and root anatomical structure of L. perenne plants.

Materials and methods

Experimental design and management

The pot experiment was carried out under controlled conditions in the phytotron chamber (Phytotron; Valašské Meziříčí, Czech Republic). Pots (15 cm in diameter and 18 cm high) were filled with Luvisol soil with a loamy sand texture (World Reference Base for Soil Resources, 2014), obtained from 20 cm of topsoil (Experimental Station Farm Field of the Warsaw University of Life Sciences, Poland, Skierniewice, 51°57′N, 20°9′E). The soil was characterized by a high content of phosphorus, potassium and magnesium (8.27, 15.77 and 7.04 mg100 g–1 of soil, respectively). It was slightly acidic (pHKCl = 5.9) with a C:N ratio of 11.03. Pre-sowing mineral fertilization was applied in the dose (g pot–1): N - 0.125, P - 0.153 and K - 0.153.

Conditions in a growth chamber were set and automatically controlled during the studies: photoperiod (day/night) 16/8 hours, radiation 95 (Wm−2) (350 PAR), the temperature increased from 12 °C at night to 26 °C in the day (2 °C per hour). The relative humidity of the air was 65%.

Lolium perenne L. turf cultivars (cvs.) Bokser and Stadion (Grunwald Plant Breeding Sp. z o.o. Grupa IHAR) were used in the tests. Eight seeds of individual cultivar were sown in each pot. Until the stress of drought, the soil moisture was maintained at 70% CWC (capillary water capacity) by watering with distilled water every day to a given weight of the soil and pot. Studies on the influence of drought on the physiological, biometric parameters and anatomical characteristics of L. perenne cvs. were carried out under two levels of soil moisture: (i) control–well-watered plants (70% CWC) and (ii) drought caused by the cessation of watering. Soil moisture determined 7, 14 and 21 days after treatment (DAT) of drought was respectively: 63.6, 40.5 and 34% CWC. Plants were exposed to drought in tillering phase (from 45th day after sowing) for 21 days. After this time, all plants were provided with optimal soil moisture conditions for the next 14 days (recovery period).

The research was performed at two levels of Si application: (i) no application of silicon (Si−) and (ii) application of silicon (Si+). The Si source was the liquid biostimulant Optysil (Intermag, Ltd., Olkusz, Poland) containing 200 g SiO2 and 24 g Fe in one liter of product in the form of sodium metasilicate (Na2SiO3) and iron chelate (Fe-EDTA). The stimulator with silicon was sprayed in the dose of 2.5 mL of solution (in the dilution 0.5 L stimulator 200 L–1 H2O per ha) directly onto the leaf blades, in four terms: 7 days before drought application, on the day of starting the stress, 7 and 14 DAT. A foil was placed between the plants in the pot to prevent the spray from reaching the soil. The tests were performed in four variants of water and Si conditions: control Si−, control Si+, drought Si−, and drought Si+.

Measurements of physiological parameters as well as biometric features of L. perenne plants were evaluated in four terms: 7, 14 and 21 DAT (period of drought) as well as 35 DAT (14 days after withholding of stress application, after recovery). The anatomical observations of plant material under the light microscope were performed in two terms: 7 and 35 DAT.

Physiological measurements

Measurements of selected chlorophyll a fluorescence parameters: the quantum efficiency of photosystem II (ΦPSII), the maximal fluorescence signal (Fm′) and steady-state (Fs) chlorophyll fluorescence yields of light-adapted samples using fluorometer FMS-2 (Modulated Chlorophyll Fluorescence System-Hansatech Instruments Ltd., Pentney, UK) were done. Chlorophyll fluorescence parameters are shown as relative units. On each plant, measurements were taken in the middle part of the fully-grown leaf blades. The saturating light source used was a built-in halogen lamp (pulse intensity equal to 8,000 μmol m−2 s−1, pulse duration 1 s). The photosynthetic electron transport rate (ETR) was calculated according to Ralph, Macinnis-Ng & Frankart (2005) as:

(1) ETR=Y×PAR×0.5×0.84

where: Y represents the quantum yield of photosystem II (PSII) electron transport (ΦPSII); 0.5—the light energy absorbed by a photosynthetic system can be distributed to the photosynthetic system PSI and PSII by the same proportion, i.e., 50% each; 0.84 - the absorption coefficient meaning that the leaves can only absorb 84% of the light energy incidence. Using a fluorometer, PAR (photosynthetically active radiation) was automatically measured.

The relative water content (RWC, %) of the L. perenne leaves was calculated from the equation (Turner, 1981):

(2) RWC(%)=freshmass–drymass/(turgidmass–drymass)×100%.

The relative content of chlorophyll (Chl) and flavonols (Flv) of plant leaves as well as nitrogen balance index (NBI) which is the ratio of Chl to epidermal Flv leaf content, were determined using hand-held leaf-clip sensor Dualex Scientific+TM (FORCE-A; Orsay, France) (Cerovic et al., 2012). Dualex units were used to express the values of measured parameters, determined in all plants, in the middle part of fully developed leaf. Measurements were performed according to the standard protocol recommended by the manufacturer.

Biometric parameters of plants

The number of shoots (SN; pcs.) on each plant of L. perenne from all variants of water and Si conditions was determined at the designated measurement terms. In order to determine the shoot dry mass (SDM; g plant–1) plants were cut at the tillering nodes level. Then shoots were dried for 72 hours at 105 °C and weighed (Turner, 1981). The soil with the roots was sieved through sieves with different mesh diameters (3–0.3 mm). The separated roots were washed under running water, cleaned of organic parts, and then scanned with an optical scanner (Epson Perfection V700 Photo, resolution of 400 dpi) and analyzed with the WinRhizo program (Regent Instruments Inc., Quebec, Canada). The root length (RL; m) and root area (RA; cm2) per plant, as well as the average diameter (RD; mm) of the roots, were measured. To determine the root dry mass (RDM; g plant–1) the root material was dried at 105 °C for 72 hours and weighed (Böhm, 1979). The ratio of the root mass to the shoot mass (R:S) was also calculated (Monk, 1966). The specific root length (SRL; m g–1) was calculated based on the data on the length of the roots and their mass (Ostonen et al., 2007):

(3) SRL=total root length/dry mass of roots.

Leaf and root anatomy

Small leaf and root fragments (2 × 3 mm and 4 × 5 mm long taking 3 cm from the apical meristem, respectively) were fixed in Karnovsky (1965) medium for 2 hours at room temperature, then washed twice in 0.1 M cacodylate buffer and postfixed for 2 hours in 1% osmium tetroxide. Next, samples were dehydrated in ethanol series (10, 20, …100%, 20 min each step) and propylene oxide (100%, for 30 minutes) and embedded in epoxy resin (SERVA, Heidelberg, Germany). Samples were sectioned using microtome Jung RM 2065. Semi-thin leaf and root cross-sections (ca. 3 µm thick) were stained with an aqueous mixture of 1% methylene blue and 1% azure A and examined under Olympus-Provis light microscope according to Borucki & Sujkowska (2008).

Statistical analysis

There was complete randomization in the design of the experiment. Three replicates for each treatment and term of measurement were used. We planned a complete set of repetitions of each factor variant for all four terms (3 replications × 2 cultivars × 2 variants of Si × 2 variants of water conditions). Measurements and observations were carried out on eight plants in the pot (n = 24). The results were statistically analyzed by multi-factor analysis of variance using the Statistica 13.3 software (Statsoft, Inc., Tulsa, OK, USA). The ANOVA assumptions of equality of variances were checked using the Levene’s test, and the normality of distribution was checked using the Q-Q test. The significance of differences was verified by Tukey’s test at significance levels P ≤ 0.05 and 0.01. The relationships between the examined features and soil moisture conditions and silicon fertilization were determined on the basis of principal components analysis (PCA).

Results

Influence of drought and Si application on physiological and biometric parameters

In the studies, the influence of water conditions and silicon application on the values of the analyzed features differed for the tested cultivars (Table 1). Our studies showed that L. perenne cvs. differed significantly in terms of Fm′, ETR, NBI and Chl. The Stadion cv was characterized by higher Fm′ and ETR values (by 7.2% and 6.8%, respectively), while the Bokser had a higher relative content of Chl and NBI (by 3.2% and 7.3%, respectively) (Table 1). Regardless of other factors, drought significantly reduced the values of almost all the tested features. The highest difference was found for ETR, Fm′ and ΦPSII compared to the control conditions (114%, 85.5% and 70.1%, respectively). The application of Si significantly increased the values of ΦPSII, ETR, NBI and RWC (by 2.3%, 3.8%, 6.3% and 4.9%, respectively) and also decreased by 10.9% the Flv values. The physiological condition of the plants also changed depending on the measurement term. Both cvs were characterized by lower values of most of the tested parameters in the following days of drought (except Flv) (Figs. 1 and 2; Tables S1 and S2). Simultaneously, providing plants with optimal soil moisture after the end of the drought resulted in a significant increase in the values of Fm′, ΦPSII, ETR and RWC (35 DAT).

Table 1 Significance of the effect of main factors on the analyzed variables based on the analysis of variance.

Mean values of the studied features.

	Fs	Fm′	ΦPSII	ETR	NBI	Chl	Flv	RWC	SN	SDM	RDM	RD	RL	RA	SRL	R:S ratio	
Cultivars (A)	ns	**	ns	**	**	*	ns	ns	**	**	ns	**	**	**	ns	*	
Bokser	649	1726a	0.526	58.9a	43.9b	28.9b	0.706	71.1	22.5b	0.24b	0.26	0.41b	25.5b	299b	79.2	1.22a	
Stadion	668	1850b	0.533	62.9b	40.9a	28.0a	0.707	72.0	16.6a	0.22a	0.25	0.39a	21.1a	259a	77.0	1.28b	
Water conditions (B)	**	**	**	**	**	**	**	**	**	**	**	ns	**	**	**	**	
Control	787b	2323b	0.667b	83.0b	51.7b	32.9b	0.692a	87.4b	28.0b	0.39b	0.43b	0.4	40.6b	482b	82.8b	0.99a	
Drought	530a	1252a	0.392a	38.8a	33.2a	24.1a	0.721b	55.7a	11.0a	0.06a	0.08a	0.4	6.0a	76a	73.4a	1.51b	
Si application (C)	ns	ns	**	**	**	ns	*	**	ns	**	**	**	**	*	ns	**	
Si–	668	1782	0.524a	59.7a	41.1a	28.2	0.743b	69.8a	19.9	0.22a	0.24a	0.41b	22.8a	272a	77.5	1.17a	
Si+	650	1793	0.536b	62.0b	43.7b	28.7	0.670a	73.2b	19.2	0.23b	0.27b	0.39a	23.9b	286b	78.6	1.33b	
DAT (D)	**	**	**	**	**	**	**	**	**	**	**	**	**	**	**	**	
7	649b	2131b	0.686d	79.0d	62.4d	40.7d	0.660a	88.0c	12.3a	0.06a	0.06a	0.47c	3.7a	55a	63.9a	0.94a	
14	613ab	1440a	0.437b	52.7b	52.5c	34.3c	0.734bc	69.5b	16.6b	0.12b	0.11b	0.36a	8.2b	94b	73.9b	0.94a	
21	597a	1423a	0.341a	41.7a	30.8b	22.4b	0.746c	58.5a	21.6c	0.32c	0.20c	0.37a	16.0c	176c	80.6c	1.46b	
35	775c	2157b	0.655c	70.1c	24.0a	16.5a	0.686ab	70.1b	27.7d	0.40d	0.65d	0.40b	65.3d	792d	93.9d	1.66c	
Notes:

The mean values of features in columns marked with the same lower-case letters did not differ significantly at *p ≤ 0.05, **p ≤ 0.01; ns–not significant.

Fs, steady-state chlorophyll fluorescence yields (relative units); Fm′, maximal fluorescence signal (relative units); ΦPSII, quantum efficiency of photosystem II (relative units); ETR, photosynthetic electron transport rate (μmol m–2 s–1); NBI, nitrogen balance index (Dualex units); Chl, content of chlorophyll (Dualex units); Flv, content of flavonols (Dualex units); RWC, relative water content (%); SN, number of shoots (pcs.); SDM, shoot dry mass (g plant–1); RDM, root dry mass (g plant–1); RD, average diameter (mm); RL, root length (m); RA, root area (cm2); SRL, specific root length (m g–1); R:S, ratio of the root mass to the shoot mass.

Figure 1 Variations in physiological features of L. perenne cultivars under various water and Si conditions after 7, 14, 21 and 35 DAT: (A and B) Fs, (C and D) Fm′, (E and F) ΦPSII, (G and H) ETR. Values show mean ± SD.

cultivars: Bokser and Stadion; water and Si conditions: control Si−, control Si+, drought Si−, drought Si+; features: Fs, steady-state chlorophyll fluorescence yields; Fm′, maximal fluorescence signal; ΦPSII, quantum efficiency of photosystem II; ETR, photosynthetic electron transport rate.

Figure 2 Variations in physiological features of L. perenne cultivars under various water and Si conditions after 7, 14, 21 and 35 DAT: (A and B) NBI, (C and D) Chl, (E and F) Flv, (G and H) RWC. Values show mean ± SD.

cultivars: Bokser and Stadion; water and Si conditions: control Si−, control Si+, drought Si−, drought Si+; features: NBI, nitrogen balance index; Chl, content of chlorophyll; Flv, content of flavonols; RWC, relative water content.

The application of Si under optimal soil moisture conditions did not significantly affect the values of chlorophyll fluorescence and RWC. In the case of the relative content of Chl and Flv, it was shown, that their values decreased under the influence of Si (Figs. 2C–2F). Under drought conditions, Si slightly increased the values of the tested parameters, especially at 14 DAT (Table S2). Only in the case of Flv content both L. perenne cultivars were characterized by lower values of this parameter.

It was found that, regardless of other factors, L. perenne cvs. were characterized by similar values of RDM and SRL (Table 1). In the case of other features, significant differences between cvs. were demonstrated. Drought reduced the values of all morphometric parameters, except RD. The use of Si, regardless of other factors, significantly increased SDM, RDM, RL, RA and R:S ratio (by 4.5, 12.5, 4.8, 5.1 and 13.8, respectively). There was no effect of Si on SN and SRL, while a 5.1% reduction in RD values after Si application was demonstrated. The application of Si in drought did not significantly differentiate most of the morphometric features (Figs. 3 and 4). However, a considerable increase in the value of R:S ratio was found (Figs. 4G and 4H; Table S2). In the case of the Bokser cv, the values increased by 30.2% and in the Stadion cultivar by 23.1%.

Figure 3 Variations in morphometric features of L. perenne cultivars under various water and Si conditions after 7, 14, 21 and 35 DAT: (A and B) SN, (C and D) SDM, (E and F) RDM, (G and H) RD. Values show mean ± SD.

cultivars: Bokser and Stadion; water and Si conditions: control Si−, control Si+, drought Si−, drought Si+; features: SN, number of shoots; SDM, shoot dry mass; RDM, root dry mass; RD, average diameter.

Figure 4 Variations in morphometric features of L. perenne cultivars under various water and Si conditions after 7, 14, 21 and 35 DAT: (A and B) RL, (C and D) RA, (E and F) SRL, (G and H) R:S. Values show mean ± SD.

cultivars: Bokser and Stadion; water and Si conditions: control Si−, control Si+, drought Si−, drought Si+; features: RL, root length; RA, root area; SRL, specific root length; R:S, ratio of the root mass to the shoot mass.

The studies showed that the values of most of the tested parameters (except RD, R:S and Flv) under drought conditions were lower compared to the control (Fig. 5). After a regeneration period under optimal soil moisture conditions (35 DAT), high values (close to the control) were recorded for ΦPSII, NBI, R:S and SRL in both cvs., especially after Si application.

Figure 5 Variability of the values of features after drought and recovery depending on the application of silicon (% in relation to corresponding variant of the control) for cultivars (A) Bokser and (B) Stadion.

Abbreviations: Fs, steady-state chlorophyll fluorescence yields; Fm′, maximal fluorescence signal; ΦPSII, quantum efficiency of photosystem II; ETR, photosynthetic electron transport rate; NBI, nitrogen balance index; Chl, content of chlorophyll; Flv, content of flavonols; RWC, relative water content; SN, number of shoots; SDM, shoot dry mass; RDM, root dry mass; RD, average diameter; RL, root length; RA, root area; SRL, specific root length; R:S, ratio of the root mass to the shoot mass.

According to PCA, the first component explained 77.90% of the examined variability under control conditions, whereas the second component explained 8.98% (Figs. 6A and 6B). At optimal soil moisture (70% CWC), silicon application and measurement term influenced both morphometric (except RD) and physiological features. A strong positive correlation was found between the morphometric parameters of roots (RDM, RA and RL) and above-ground parts of plants (SN and SDM) and between physiological parameters (ETR, RWC and NBI). However, a strong negative relationship under control conditions was demonstrated between Fs and ΦPSII and Chl and between NBI and roots features: RDM, RA and RL.

Figure 6 Results of principal component analysis (PCA) presenting relationships between all analyzed parameters (A) in control and (C) in drought and for the conditions of growth (B) in control and (D) drought.

Abbreviations: silicon application: lack of silicon–Si− and silicon application–Si+; days after treatment: 7, 14, 21 and 35 DAT. Fs, steady-state chlorophyll fluorescence yields; Fm′, maximal fluorescence signal; ΦPSII, quantum efficiency of photosystem II; ETR, photosynthetic electron transport rate; NBI, nitrogen balance index; Chl, content of chlorophyll; Flv, content of flavonols; RWC, relative water content; SN, number of shoots; SDM, shoot dry mass; RDM, root dry mass; RD, average diameter; RL, root length; RA, root area; SRL, specific root length; R:S, ratio of the root mass to the shoot mass.

The PCA analysis carried out for drought conditions showed that the first component was responsible for 50.54% and the second for 33.79% of the analyzed variability (Figs. 6C and 6D). Drought had the strongest impact on RWC. Positive relationships were found between RL, RDM and RA and between ΦPSII, Fm′ and ETR. Moreover, a negative correlation was found between R:S and Chl and NBI. The NBI and Chl parameters reached the highest values at 7 DAT, while after the regeneration period (35 DAT) the highest values were found in the case of RL, RDM and RA.

Influence of drought and Si application on leaf and root anatomy

There were differences in the anatomical characteristics of plant leaves according to water conditions and Si application. Generally, in the case of both cvs, Si application during drought, compared to drought without Si, increased the leaves thickness, the height of the ridges and reduced the width and distance between them (Table 2). This reduced distance between the ridges in Si-treated leaves allows for greater bulliform cell depth in the upper epidermis compared to drought. These cells are contribute to water loss reduction via the curling leaf blade. The same relationships were observed after the regeneration period.

Table 2 Main anatomical characters observed in the leaf cross-sections of L. perenne cultivars under various water and Si conditions after the 7 DAT (drought) and 35 DAT (recovery).

Cultivar	Water and Si conditions	Characters	
high of the ridges in the leaf (µm)	width of the ridges (µm)	distance between the ridges (µm)	
		7 DAT (drought)	
Bokser	Control Si−	263.8 ± 33.3	336.5 ± 22.2	58.9 ± 7.5	
Drought Si−	237.1 ± 34.8	245.8 ± 46.2	77.7 ± 16.6	
Control Si+	283.8 ± 17.0	220.4 ± 18.2	32.1 ± 5.3	
Drought Si+	278.8 ± 53.8	216.6 ± 14.2	42.9 ± 2.9	
Stadion	Control Si−	305.5 ± 35.0	250.3 ± 35.6	73.5 ± 13.0	
Drought Si−	232.8 ± 48.5	262.4 ± 29.2	56.0 ± 8.1	
Control Si+	357.3 ± 18.9	236.3 ± 40.0	67.3 ± 8.0	
Drought Si+	277.0 ± 10.4	204.9 ± 36.4	55.1 ± 8.4	
		35 DAT (recovery)	
Bokser	Control Si−	266.3 ± 34.1	230.5 ± 62.7	51.5 ± 5.1	
Drought Si−	236.7 ± 10.9	233.7 ± 2.0	43.4 ± 1.1	
Control Si+	295.3 ± 52.7	256.4 ± 75.0	39.1 ± 5.8	
Drought Si+	260.0 ± 21.2	226.9 ± 2.5	33.3 ± 2.0	
Stadion	Control Si−	283.4 ± 29.6	233.7 ± 18.1	52.1 ± 2.6	
Drought Si−	231.4 ± 4.9	241.3 ± 9.5	41.2 ± 3.7	
Control Si+	349.6 ± 18.9	196.6 ± 59.9	18.9 ± 5.6	
Drought Si+	270.0 ± 5.9	204.9 ± 36.4	19.2 ± 3.5	
Note:

Cultivars, Bokser and Stadion; water conditions, control and drought; silicon application, lack of silicon–Si− and silicon application–Si+.

The adaxial surface of the leaves was corrugated (Fig. 7). The upper (adaxial) epidermis contains stomata and bulliform (motor) cells. Independently on water and Si conditions, mesophyll cells were usually spherical in shape. Mesophyll cells, adjacent to the lower epidermis developed as a tightly packed cell layer. External cell walls of the upper and lower epidermis were better developed under drought conditions than in the control. Moreover, Si application stimulated the development of the external cell walls of both epidermis.

Figure 7 Effects of water conditions and Si application on the anatomy of the L. perenne leaves and roots (left and right column for each cultivar, respectively) after 7 DAT (drought).

Abbreviations: cultivars: Bokser and Stadion; water conditions: control and drought; silicon application: lack of silicon–Si− and silicon application–Si+. Light microscopy. Cross-sections were taken through the central parts of the newly emergin leaf-blades and ca. 3 cm below root apexes of newly emerging roots. Abbreviations: bc, bulliform cells; ue, upper (adaxial) epidermis; le, lower (abaxial) epidermis; st, stomata; m, mesophyll; s, sclerenchyma; ss, substomatal space; vb, vascular boundle; en, endodermis; ex, exodermis; r, rhizoderm; l, lacuna; mk, metaxylem vessel; vc, vascular cylinder; Scale bars = 50 µm.

In the roots, we observed exoderm development regardless of the water conditions and Si application. The second subepidermal cell layer developed as an exoderm independent of water conditions, Si application, and plant cvs. (Stadion or Bokser). In the case of plants in control without Si application (control Si‒), exodermal cells exhibited full turgor. In contrast to the lack of Si (Si‒), its application (Si+) resulted in the maintenance of turgor in exodermal cells of both cultivars under drought conditions. Furthermore, Si presence often resulted in the development of 2-layered exodermis under drought conditions in the case of Stadion cv (Fig. 7). In the root cortex, characteristic lacunas were formed by the disintegration of superordinately arranged cells in the radial direction. Lacunas developed in the root cortex regardless of the treatment and cultivar. One or two cortical cell layers adjacent to endodermis developed as sclerenchymatic cells, which, independently of treatment, did not participate in lacunas formation. After the recovery period (35 DAT), in the case of both cultivars, Si application resulted in better development of root lacunas both in control and drought conditions (Fig. 8).

Figure 8 Effects of water conditions and Si application on the anatomy of the L. perenne leaves and roots (left and right column for each cultivar, respectively) after 35 DAT (recovery).

Abbreviations: cultivars: Bokser and Stadion; water conditions: control and drought; silicon application: lack of silicon–Si− and silicon application–Si+. Light microscopy. Cross-sections were taken through the central parts of the newly emergin leaf-blades and ca. 3 cm below root apexes of newly emerging roots. Abbreviations: bc, bulliform cells; ue, upper (adaxial) epidermis; le, lower (abaxial) epidermis; st, stomata; m, mesophyll; s, sclerenchyma; ss, substomatal space; vb, vascular boundle; en, endodermis; ex, exodermis; r, rhizoderm; l, lacuna; mk, metaxylem vessel; vc, vascular cylinder; Scale bars = 50 µm.

Discussion

In the case of turf grasses, the main goal of cultivation is to provide an aesthetic aspect and good soil cover with plants. Therefore, the ability of plants to maintain healthy and good condition of both leaves and roots is crucial, especially under stress. We investigated the effects of drought stress under silicon application conditions on the physiological and morphometric features of L. perenne cultivars as well as the anatomy of the leaves and roots of the plants.

Plants tolerate drought by suspending the growth of their shoots (Fry & Huang, 2004). They often become dormant. The stem apex and leaf primordia endure under such circumstances, but the leaves may desiccate and die (Seleiman et al., 2021). When sufficient rainfall and favorable growing conditions are restored, the plants recover. In our studies, drought lasting 21 days inhibited the growth of both grass shoots and roots. This was confirmed by low values of biometric parameters on subsequent days of drought (SN, SDM, RDM, RL and RA) and decreasing values of physiological parameters (especially Fm′, ΦPSII and ETR). Plants under drought stress produce less dry matter of roots due to adaptation mechanisms meant to prevent excessive water loss (Mastalerczuk & Borawska-Jarmułowicz, 2021). It also allows the reduction of the adverse effect of drought on the physiological metabolism of plants, especially their photosynthetic activity (Fariaszewska et al., 2020). Closing the stomata due to drought reduces the efficiency of the CO2 assimilation reaction and also reduces the consumption of ATP and NADPH (Dias & Brüggemann, 2010). As a result, in our studies, electron transport (ETR) in the light phase slowed down and caused a decrease in the ΦPSII value.

Harvesting light energy to drive the electron transport reactions in the photosynthesis process is closely related to the chlorophyll content in the leaves (Croft et al., 2017). According to the literature, the degree of chlorophyll degradation caused by a moisture deficit can be linked to stress sensitivity (Chauhan et al., 2023). In the case of drought-sensitive genotypes, components of the photosynthetic apparatus may be damaged, whereas drought-tolerant genotypes may show good adaptability to reduce/avoid drought stress disturbances (RongHua et al., 2006; Staniak et al., 2020; Mastalerczuk & Borawska-Jarmułowicz, 2021). Our research showed that the Chl content in L. perenne leaves decreased on subsequent days of drought, which could indicate chlorophyll metabolism disorders. One of the crucial processes of drought adaptation is the buildup of substances that boost antioxidant capacity and support the detoxification of reactive oxygen species (Hasanuzzaman et al., 2020). An important role in the protection of the photosynthetic apparatus is played by non-enzymatic flavonoids, which are preferentially located in the epidermal cells of leaves (Agati et al., 2012; Tan & Gören, 2024). In our study, increased flavonol accumulation was evident throughout the stress period (up to 21 DAT). After providing the plants with optimal soil moisture after a period of drought (35 DAT), a significant increase in the values of chlorophyll fluorescence parameters (Fm′, ΦPSII, ETR) and RWC and also a decrease in relative Flv content was demonstrated, which may indicate the tolerance of L. perenne to drought conditions. Drought independently on treatment and variety resulted in thicker outer cell wall development of the upper and lower leaf epidermis, which may also give better resistance to water evaporation.

In many studies, the positive effects of silicon were observed almost exclusively under stressful conditions (Coskun et al., 2019). Similarly in our research, Si supplementation under control conditions did not differentiate the values of chlorophyll fluorescence parameters and RWC. However, it limited the relative content of Chl and Flv in L. perenne plants. Due to their antioxidative properties, flavones protect the plant against reactive oxygen species (ROS) (Mierziak, Kostyn & Kulma, 2014). According to our earlier research, plants with a very dry and moderately moist year had a higher relative flavonoid content than those with a rainy year. In that moisture condition, the application of Si caused a decrease in flavone content, especially in L. perenne plants (Mastalerczuk et al., 2020).

Several studies have demonstrated that when Si was applied to drought-stressed plant species, the concentration of photosynthetic pigments increased. In the studies of Shen et al. (2010), Si application under drought increased total chlorophyll contents in soybean leaves, while Yin et al. (2014) reported a similar reaction in the case of sorghum. The high levels of Si in plant leaves promote the biosynthesis of chlorophyll pigments and slow down their breakdown, which is caused by water stress (Verma et al., 2022). According to Rizwan et al. (2015), Si application in drought causes changes in values of gas exchange parameters, water potential, and reduction in oxidative stress in plants. Additionally, it can improve gas exchange, which has a positive effect on photosynthesis, nutrient uptake, and eventually, plant biomass and growth. The beneficial effect of silicon fertilization on plants under drought conditions may be due to an improved water balance, more effective osmoregulation, and reduced water loss through transpiration (Sacała, 2009; Wang et al., 2021).

In our study, the application of Si in drought increased values of chlorophyll a fluorescence parameters (ΦPSII, ETR) and RWC. Furthermore, RWC values in plants treated with Si were higher after the recovery period (35 DAT). The higher RWC demonstrates that Si enhanced water uptake and retention in L. perenne plants under drought stress. As suggested by Gong et al. (2003) the increase in leaf water content and water potential in wheat plants under drought and Si presence may be caused by the leaves’ thickness in comparison to those without Si treatment. Additionally, since water molecules may not be able to escape from the leaf surface, the deposition of Si in leaves may decrease transpiration (Ahmed, Asif & Hassan, 2014; Keller et al., 2015).

Most plant water loss occurs through leaf transpiration through stomata. Water loss through transpiration significantly decreases as leaf area decreases (Trlica & Biondini, 1990). In the present study, we observed that Si-treated leaves were thicker, and the upper blade surface had greater protuberances compared to drought. The relative humidity of the leaf surface can be maintained by a wavy surface, thicker mesophyll and leaf layers, and structures that aid in plant water storage. Additionally, the greater depth of the bulliform cells of Si-treated leaves facilitates the rolling of the leaf blade to avoid water loss during water stress (Dickison, 2000).

Based on data from the literature, silicon is primarily found in the apoplast, where it can aid in the formation of mechanical strength and physical barriers, such as in the root endodermis or leaf epidermal cell wall (Hodson & Sangster, 1989; Dietrich et al., 2003; Luyckx et al., 2017; Coskun et al., 2019). Our studies showed that, root cortex parenchyma cells arranged in radial rows disintegrate by the disruption of the tangential walls. Because radial walls remain untouched, so-called lacunas are formed. In the literature, lacunas are described as evident features of aerenchyma formation. In both cultivars of L. perenne, drought without Si application resulted in at least partially collapsed exodermal cells, which is a sign of turgor loss (Cuneo et al., 2016). Supplementation of Si resulted in fully rounded exodermal cells, which indicates their full turgor. Microscopic observations suggest that exodermis may be crucial for efficient resistance to drought conditions, especially together with Si application, which leads to the development of two-cell-layer exodermis (cv. Stadion, Fig. 5).

According to Farooq et al. (2009), applying Si during drought stress is crucial for maintaining root development and water transport. In our earlier studies (Mastalerczuk, Borawska-Jarmułowicz & Darkalt, 2023), we found that, under the drought and Si applications, both the water use efficiency and also the length and mass of roots of the plants increased significantly compared to the conditions without silicon. Additionally, plants transferred carbon from their leaves to form fine roots (with small diameters) in response to drought. In this study, we found that, under similar conditions, not only the RWC values but also the root-to-shoot ratio increased significantly compared to the treatments without silicon, both in drought and after the recovery period. These findings show that when Si supplementation was applied, the tested plants developed roots in response to drought stress, reducing the amount of photosynthetic products allocated to the aboveground organs (leaves and stems). One explanation for this could be that plants use newly generated biomass to strengthen their roots, which can enhance their ability to access water in the soil. This is a survival strategy for plants experiencing water scarcity.

An indicator that describes the plant’s responses to modifications in the soil environment is specific root length (Ostonen et al., 2007; Liu et al., 2024). Plants demonstrating a higher share of fine roots of smaller diameter (higher value of SRL) have better root branching, allowing them to survive periods of water or nutrient deficiencies (Hill et al., 2006; Fort & Freschet, 2020). In our study, L. perenne plants subjected to drought conditions were characterized by low SRL values. Only after recovery in the presence of Si they were distinguished by the better branching of the roots than plants without Si. This indicates that Si mitigates the harmful impacts of drought conditions, accelerates root regeneration, and promotes plant development after a drought by increasing most of the studied physiological parameters.

Conclusion

The present study demonstrated that 21-day drought stress in the tillering phase of young L. perenne seedlings markedly decreased RWC, chlorophyll content and photosynthesis activity while increasing the flavonol content. These adverse impacts were reflected in the reduction of biomass-related traits: shoot number, mass of shoots and root, as well as area and length of roots. Foliar Si supplementation under drought stress conditions influenced plant growth through better electron transport efficiency, more efficient quantum yield of photochemical reactions in PSII and increased RWC and chlorophyll content, promoting photosynthesis and stimulating biomass allocation to the roots (higher R:S values). Under drought stress conditions, Si application leads to the development of two-cell-layer exodermis, which reduces the water losses in L. perenne roots and shoots and, as a result, improves the drought tolerance of plants. To maintain green areas in cities under changing climate conditions, it is essential to comprehend how drought affects turfgrass species and to develop strategies that improve plant adaptation to drought. Given the complex relationships between Si and different grass species, genotypes and environments, detailed studies are needed to understand the interactions between Si and plant responses under drought conditions.

Supplemental Information

Supplemental Information 1 Raw data of all measured parameters.

Fs – steady-state chlorophyll fluorescence yields (relative units) , Fm′ – maximal fluorescence signal (relative units), ΦPSII – quantum efficiency of photosystem II (relative units), ETR– photosynthetic electron transport rate (μmol m–2 s–1), NBI – nitrogen balance index (Dualex units), Chl – content of chlorophyll (Dualex units), Flv – content of flavonols (Dualex units), RWC – relative water content (%), SN – number of shoots (pcs.), SDM – shoot dry mass (g plant–1), RDM – root dry mass (g plant–1), RD – average diameter (mm), RL – root length (m), RA – root area (cm2), SRL – specific root length (m g–1), R:S – ratio of the root mass to the shoot mass.

Supplemental Information 2 Results of ANOVA (p-values) presenting the interactions for analyzed variables: cultivars (A), water conditions (B), Si application (C) and DAT (D).

Abbreviations: Fs – steady-state chlorophyll fluorescence yields (relative units), Fm′ – maximal fluorescence signal (relative units), ΦPSII – quantum efficiency of photosystem II (relative units), ETR – photosynthetic electron transport rate (μmol m–2 s–1), NBI – nitrogen balance index (Dualex units), Chl – content of chlorophyll (Dualex units), Flv – content of flavonols (Dualex units), RWC – relative water content (%), SN – number of shoots (pcs.), SDM – shoot dry mass (g plant–1), RDM – root dry mass (g plant–1), RD – average diameter (mm), RL – root length (m), RA – root area (cm2), SRL – specific root length (m g–1), R:S – ratio of the root mass to the shoot mass.

Supplemental Information 3 Detailed comparisons of means for individual terms (7, 14, 21 and 35 DAT).

The mean values of features in columns, for individual terms, marked with the same lower-case letters did not differ significantly at p ≤ 0.05 Abbreviations: Fs – steady-state chlorophyll fluorescence yields (relative units), Fm′ – maximal fluorescence signal (relative units), ΦPSII – quantum efficiency of photosystem II (relative units), ETR – photosynthetic electron transport rate (μmol m–2 s–1), NBI – nitrogen balance index (Dualex units), Chl – content of chlorophyll (Dualex units), Flv – content of flavonols (Dualex units), RWC – relative water content (%), SN – number of shoots (pcs.), SDM – shoot dry mass (g plant–1), RDM – root dry mass (g plant–1), RD – average diameter (mm), RL – root length (m), RA – root area (cm2), SRL – specific root length (m g–1), R:S – ratio of the root mass to the shoot mass.

Additional Information and Declarations

Competing Interests

Piotr Dąbrowski is an Academic Editor for PeerJ. The other authors declare that they have no competing interests.

Author Contributions

Grażyna Mastalerczuk conceived and designed the experiments, performed the experiments, analyzed the data, prepared figures and/or tables, authored or reviewed drafts of the article, and approved the final draft.

Barbara Borawska-Jarmułowicz performed the experiments, authored or reviewed drafts of the article, and approved the final draft.

Marzena Sujkowska-Rybkowska performed the experiments, analyzed the data, prepared figures and/or tables, authored or reviewed drafts of the article, and approved the final draft.

Magdalena Bederska-Błaszczyk performed the experiments, prepared figures and/or tables, and approved the final draft.

Wojciech Borucki performed the experiments, analyzed the data, prepared figures and/or tables, authored or reviewed drafts of the article, and approved the final draft.

Piotr Dąbrowski analyzed the data, authored or reviewed drafts of the article, and approved the final draft.

Data Availability

The following information was supplied regarding data availability:

The raw data are available in the Supplemental Files.

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
