# Peer review of "Silicon mitigates the adverse effects of drought on Lolium perenne physiological, morphometric and anatomical characters"

_PeerJ, doi:10.7717/peerj.18944_

## Round 0.1 · original submission · Major Revisions

Drought stress has emerged as one of the most significant environmental challenges for agricultural and forage crops over the past decade. Its adverse effects are becoming increasingly evident, leading to reduced crop yields, compromised forage quality, and widespread implications for food security. This phenomenon has also spurred extensive research efforts, aiming to develop adaptive strategies and improve crop resilience under such conditions. Your research has the potential to make a meaningful contribution to this growing body of knowledge by offering new insights into the mechanisms and management of drought stress.

However, to maximize the impact and clarity of your findings, it is essential to address certain technical aspects and refine your article accordingly. A thorough and thoughtful review of the reviewers' suggestions is strongly recommended. Carefully consider each recommendation, assessing its relevance and potential to enhance your work. If you find yourself disagreeing with any specific suggestion, it is important to provide clear and well-reasoned justifications, supported by evidence where applicable, to substantiate your perspective. By doing so, you will not only strengthen the scientific rigor of your study but also foster constructive dialogue with the reviewers and broader research community.

Reviewer 1 ·

Basic reporting

The manuscript is well-written and clear. A couple of minor comments:
- In the discussion, it might be worth highlighting where previous studies have used soil rather than foliar application of Si and mentioning that the underlying mechanisms may be different
- Line 39: it would be clearer to simply say “with and without Si application”; at present, the “two variants of Si application” could indicate e.g. 5 vs 10 foliar Si applications
- Line 107: please clarify that Si can be regarded as essential only for some species (particularly grasses); there are several species that are regarded as Si “excluders”
- Line 106: please replace “important” with e.g. “abundant”

Experimental design

The authors present a novel study on the effect of Si during drought stress in turfgrass. The methods section is well-written and detailed. However, I have a few comments:
- Lines 155-162: did -Si plants receive any control treatment (e.g. sprayed with water/ solution to balance the Na and Fe)?
- Line 163: were parameters measured before or after Si treatment (which seems to have been applied on the same day)?

Validity of the findings

This is the one area where the manuscript still requires some further work. No mention is made of checking the ANOVA assumptions (e.g. Levene’s test, Shapiro test, visual inspection of Q-Q plots). From the raw data, it appears that 3 plants were selected on each harvest date, but in Fig 1, it is stated that n = 24. Please could you clarify? If the same plants were measured on multiple days, then the ANOVA assumption of independence is broken. The easiest solution is probably to perform separate ANOVAs for different days, but some form of time-series analysis might also be appropriate. Additionally, an extra factor level should be added into the model to account for there likely being far less variation for the 8 plants grown in the same pot compared to plants grown in the other two pots.
The figures may also require further editing. At present, figs 1 and 2 contains a lot of data and are quite small and could easily be split into more separate figures, whereas I think figs 3 and 4 could be removed without affecting the conclusions of the study.
The diagram in table 2 is very helpful and it is clear to understand. I think table 1 is its current form is a little confusing. The interactions are mostly significant, so it’s not really justified to present the means of e.g. just the 2 cultivars, pooling together all the other variables. It might be more useful to add in the F- or p-values rather than the factor means.

Reviewer 2 ·

Basic reporting

English language should be improved in revised version. References are old which should be replaced with latest ones.

Experimental design

Experimental design is appropriate

Validity of the findings

Findings are satisfactory but poorly expressed in result section

Additional comments

File attached.

Annotated reviews are not available for download in order to protect the identity of reviewers who chose to remain anonymous.

---

## Round 0.2 · accepted · Accept

I appreciate your constructive attitude toward the reviewers' suggestions and improving your article based on their suggestions. I believe your manuscript is now ready for publication. We look forward to your next article.

Reviewer 1 ·

Basic reporting

Previous comments have now been addressed.

Experimental design

Previous comments have now been addressed.

Validity of the findings

Previous comments have now been addressed.

Reviewer 2 ·

Basic reporting

No comments

Experimental design

No comments

Validity of the findings

No comments

Additional comments

No comments